# rang: Reconstructing reproducible R computational environments

**Chung-hong Chan** **\*, David Schoch**

GESIS Leibniz-Institut für Sozialwissenschaften, Mannheim, Germany

\* chung-hong.chan@gesis.org

## Abstract

A complete declarative description of the computational environment is usually missing when researchers share their materials. Without such description, software obsolescence and missing system components can jeopardize computational reproducibility in the future, even when data and computer code are available. The R package rang is a complete solution for generating the declarative description for other researchers to automatically reconstruct the computational environment at a specific time point. The reconstruction process, based on Docker, has been tested for R code as old as 2001. The declarative description generated by rang satisfies the definition of a reproducible research compendium and can be shared as such. In this contribution, we show how rang can be used to make otherwise unexecutable code, spanning fields such as computational social science and bioinformatics, executable again. We also provide instructions on how to use rang to construct reproducible and shareable research compendia of current research. The package is currently available from CRAN (https://cran.r-project.org/web/packages/rang/index.html) and GitHub (https://github.com/chainsawriot/rang).

## Background

*"In some cases the polarization estimation will not work . . . This is* NOT *a problem in the method, it is entirely dependent on the numpy version (and even the OS's). If you have different versions of numpy or even the same version of numpy on a different OS configuration, different networks will fail randomly. . . [F]or instance, the 109th Congress will fail, but will work entirely normally on a different numpy version, which will fail on a different Congress network."*

~ excerpt of the README file of a software for polarization estimation

Other than bad programming practices [1], the main computing barrier to computational reproducibility is the failure to reconstruct the computational environment like the one used by the original researchers. This task looks trivially simple. But as computer science research has shown, this task is incredibly complex [2, 3]. For a usual scripting language such as R [4] (also a popular programming language used frequently in various computational fields, e.g. computational social science or bioinformatics) as an example, that pertains to four aspects: A) operating system, B) system components such as `libxml2`, C) the exact R version, and D) what and which version of the installed R packages. We will call them Component A, B, C, D

**Data Availability Statement:** Data and code used in this paper are available here: https://github.com/chainsawriot/rang/tree/v0.3/paper.

**Competing interests:** The authors have declared that no competing interests exist.

in the following sections. Any change in these four components can possibly affect the execution of any shared computer code. For example, the lack of the system component `libxml2` can impact whether the R package `xml2` can be installed on a Linux system. If the shared computer code requires the R package `xml2`, then the whole execution fails.

In reality, the impact of Component A is relatively weak as mainstream open source programming languages and their software libraries are usually cross platform. In modern computational research, Linux is the de-facto operating system in High Performance Computing environments. Instead, the impact of Components B, C, and D is much higher. Component D is the most volatile among them all as there are many possible combinations of R packages and versions. Software updates with breaking changes (even in a dependency) might render existing shared code using those changed features not executable or not producing the same result anymore. Also, software obsolescence is commonplace, especially since academic software is often not well maintained due to lack of incentives [5].

The DevOps (software development and IT operations) community is also confronted with this problem. The issue is usually half-jokingly referred to as "it works on my machine"-problem [6, a software works on someone's local machine but is not working anymore when deployed to the production system, indicates the software tacitly depends on the computational environment of the local machine]. A partial solution to this problem from the DevOps community is called *containerization*. In essence, to containerize is to develop and deploy the software together with all the libraries and the operating system in an OS-level virtualization environment. In this way, software dependency issues can be resolved inside the isolated virtualized software environment and independent of what is installed on the local computer. Docker is a popular choice in the DevOps world for containerization.

To build a container, one needs to write a plain text declarative description of the required computational environment. Inside this declarative description, it should unambiguously specify all four Components mentioned above. For Docker, it is in the form of a plain text file called `Dockerfile`. This `Dockerfile` is then used as the recipe to build a Docker image, where the four Components are assembled. Then, one can launch a container with the built Docker image.

There has been many papers written on how containerization solutions such as Docker can be helpful also to foster computational reproducibility of science [e.g. 7–9]. Although tutorials are available [e.g. 7], providing a declarative description of the computational environment in the form of `Dockerfile` is far from the standard code sharing practice. This might be due to a lack of (DevOps) skills of most scientists to create a `Dockerfile` [10]. But there are many tools available to automate this process [e.g. 7]. The case in point described in this paper, `rang`, is one of them. We argue that `rang` is the only easy-to-use solution available that can specify and restore all four components without the reliance on any commercial service.

## Existing solutions

`renv` [11] (and its derivatives such as `jetpack` and its predecessor `packrat`) takes a similar approach to Python's `virtualenv` and Ruby's `Gem` to unambiguously specify the exact version of R packages using a "lock file". Other solutions such as `checkpoint` [12] depend on the availability of The Microsoft R Application Network (MRAN, a time-stamped daily backup of CRAN), which will be shut down on July 1, 2023. `groundhog` [13] used to depend on MRAN but has a plan to switch to their home-grown R package repository. These solution can effectively specify Component C and D. But they can only restore component D. Also, for solutions depending on MRAN, there is a limit on how far back this reproducibility can go, since MRAN can only go back as far as September 17, 2014. Additionally, it only covers CRAN packages.

`containerit` [7] takes the current state of the computational environment and documents it as a `Dockerfile`. `containerit` makes the assumption that Component A has a weak influence on computational reproducibility and therefore defaults to Linux-based Rocker base images [9]. In this way, it fixes Component A. But `containerit` does not specify the exact version of R packages. Therefore, it can specify components A, B, C, but only a part of component D. `dockta` is another containerization solution that can potentially specify all components due to the fact that MRAN is used. But it also suffers from the same limitations of MRAN mentioned above, i.e. shutting down on July 1, 2023, going back as far as September 17, 2014, and covering only CRAN.

It is also worth mentioning that MRAN is not the only archival service. Posit also provides a free (*gratis*) time-stamped daily backup of CRAN and Bioconductor (a series of repositories of R package for bioinformatics and computational biology) called Posit Public Package Manager. It can go as far back as October 10, 2017.

These solutions are better for prospective usage, i.e. using them now to ensure the reproducibility of the current research for future researchers. `rang` mostly targets retrospective usage, i.e., to reconstruct historical R computational environments for which the declarative descriptions are not available. One can think of `rang` as an archaeological tool. In this realm, we could not find any existing solution targeting R specifically which does not currently depend on MRAN.

## Basic usage

There are two important functions of `rang` version 0.2.0: `resolve()` and `dockerize()`.

`resolve()` queries various web services from the r-hub project of the R Consortium for information about R packages at a specific time point that is necessary for reconstructing a computational environment, e.g. (deep) dependencies (Component D), R version (Component C), and system requirements (Component B). For instance, if there was a computational environment constructed on 2020-01-16 (called "snapshot date") with the several natural language processing R packages, `resolve()` can be used to resolve all the dependencies of these R packages. Currently, `rang` supports CRAN, Bioconductor, GitHub, and local packages.

```
library(rang)
graph <- resolve(pkgs = c("openNLP", "LDAvis", "topicmodels", "quanteda"),
                 snapshot_date = "2020-01-16")
graph
```

The resolved result is an S3 object called `rang`. The information contained in a `rang` object can then be used to construct a computational environment in a similar manner as `containerit`, but with the packages and R versions pinned on the snapshot date. Then, the function `dockerize()` is used to generate the `Dockerfile` and other scripts in the `output_dir`.

```
dockerize(graph, output_dir = "docker")
```

For R >= 3.1, the base images from the Rocker project are used [9]. For R < 3.1 but >= 1.3.1, a custom base image based on Debian is used. As of writing, `rang` does not support R < 1.3.1, i.e. snapshot dates earlier than 2001-08-31 (which is 13 years earlier than all solutions depending on MRAN). There are two features of `dockerize()` that are important for future reproducibility.

1. By default, the Docker image building process downloads source packages from their sources and then compiles them. This step depends on the future availability of R packages on CRAN (which is extremely likely to be the case in the near future, given the continuous availability since 1997-04-23), Bioconductor, and Github. However, it is also possible to cache (or archive) the source packages now. The archived R packages can then be used during the building process instead. The significance of this step in terms of long-term computational reproducibility will be discussed in the section on executable compendia.

```
dockerize(graph, output_dir = "docker", cache = TRUE)
```

2. It is also possible to install R packages in a separate library during the building process to isolate all these R packages from the main library.

```
dockerize(graph, output_dir = "docker", cache = TRUE,
          lib = "anotherlibrary")
```

For the sake of completeness, the instructions for building the Docker image and running the Docker container on Unix-like systems are included here.

```
cd docker
## might need to sudo
docker build -t rangimg.
## interactive environment
docker run --rm --name "rangcontainer" -ti rangimg
```

### Project scanning

The first argument of `resolve()` is processed by a separate function called `as_pkgrefs()`. For interoperability, `rang` supports the "package references" standard used also in other packages such as `renv` [11]. It is mostly used for converting "shorthands" (e.g. `xml2` and `S4Vectors`) to package references (e.g. `cran::xml2` and `bioc::S4Vectors`).

When `as_pkgrefs()` is applied to a single path of a directory, it scans all relevant files (`DESCRIPTION`, R scripts and R Markdown files) for all R packages used (based on `renv::dependencies()`). Its functionality is demonstrated in three of the following examples below. However, an important caveat is that it can only scan CRAN and Bioconductor packages.

## Case studies

In this section, we present several examples for how `rang` can be used to make shared, but otherwise unexecutable, R code runnable again. The examples were drawn from various fields spanning from political science, psychological science, and bioinformatics.

### quanteda JOSS paper

The software paper of the text analysis R package `quanteda` was published on 2018-10-06 [14]. In the paper, the following R code snippet is included.

```
library("quanteda")
# construct the feature co-occurrence matrix
examplefcm <-
tokens(data_corpus_irishbudget2010, remove_punct = TRUE) %>%
tokens_tolower() %>%
tokens_remove(stopwords("english"), padding = FALSE) %>%
fcm(context = "window", window = 5, tri = FALSE)
# choose 30 most frequency features
topfeats <- names(topfeatures(examplefcm, 30))
# select the top 30 features only, plot the network
set.seed(100)
textplot_network(fcm_select(examplefcm, topfeats), min_freq = 0.8)
```

On 2023-02-08, this code snippet is not executable with the current version of `quanteda` (3.2.4). It is possible to install the "period appropriate" version of `quanteda` (1.3.4) using `remotes` on the current version of R (4.2.2). And indeed, the above code snippet can still be executed.

```
remotes::install_version("quanteda", version = "1.3.4")
```

The problem is that installing `quanteda` 1.3.4 this way installs the latest dependencies from CRAN and not the "period appropriate" ones. `quanteda` 1.3.4 uses a deprecated (but not yet removed) function of `Matrix` (`as(<dgTMatrix>, "dgCMatrix")`). If this function is removed in the future, the above code snippet will not be executable anymore.

Using `rang`, one can query the version of `quanteda` on 2018-10-06 and create a Docker container with all the "period appropriate" dependencies. Here, the `rstudio` Rocker image is selected.

```
library(rang)
graph <- resolve(pkgs = "quanteda",
                 snapshot_date = "2018-10-06",
                 os = "ubuntu-18.04")
dockerize(graph, output_dir = "quanteda_docker",
          image = "rstudio")
```

The above code snippet can be executed with the generated container without any problem Fig 1.

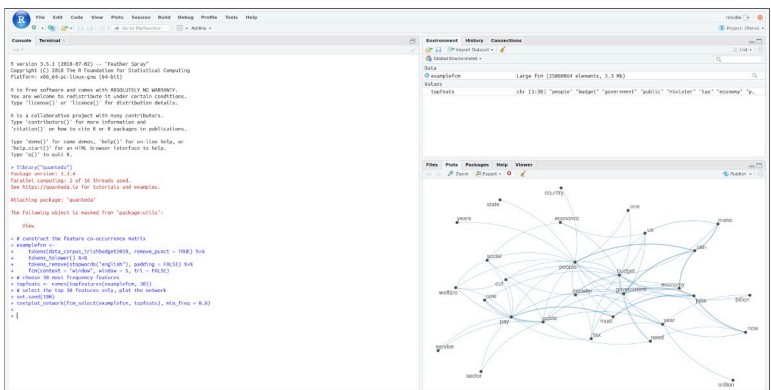

**Fig 1. The code snippet running in a R 3.5.1 container created with rang.**

## Psychological science

Crüwell et al. [15] evaluate the computational reproducibility of 14 articles published in *Psyhocological Science*. Among these articles, the paper by Hilgard et al. [16] has been rated as having "package dependency issues".

All data and computer code are available from GitHub with the last commit on 2019-01-17. The R code contains a list of R packages used in the project as `library()` statements, including an R package on GitHub that is written by the main author of that paper. However, we identified one package (`compute.es`) that was not written in those `library()` statements but used with the namespace operator, i.e. `compute.es::tes()`. This undocumented package can be detected by `renv::dependencies()`, of which `as_pkgrefs()` is a wrapper.

Based on the above information, one can run `resolve()` to obtain the dependency graph of all R packages on 2019-01-17.

```
## scan all packages
r_pkgs <- as_pkgrefs("vvg-2d4d")
## replace cran::hilgard with Github
r_pkgs[r_pkgs == "cran::hilhard"] <- "Joe-Hilgard/hilgard"
graph <- resolve(r_pkgs, snapshot_date = "2019-01-17")
```

When running `dockerize()`, one can take advantage of the `materials_dir` parameter to transfer the shared materials from Hilgard et al. [16] into the Docker image.

```
dockerize(graph, "hilgard", materials_dir = "vvg-2d4d", cache = TRUE)
```

We then built the Docker and launch a Docker container. For this container, we changed the entry point from R to bash so that the container goes to the Linux command shell instead.

```
cd hilgard
docker build -t hilgard .
docker run --rm --name "hilgardcontainer" --entrypoint bash -ti hilgard
```

Inside the container, the material is located in the `materials` directory. We used the following shell script to test the reproducibility of all R scripts.

```
cd materials
rfiles=(0_data_aggregation.R 1_data_cleaning.R 2_analysis.R 3_plotting.R)
for i in ${rfiles[@]}
do
   Rscript $i
   code=$?
   if [$code != 0]
   then
       exit 1
   fi
done
```

All R scripts ran fine inside the container and the figures generated are the same as the ones in Hilgard et al. [16].

## Political analysis

The study by Trisovic et al. [1] evaluates the reproducibility of R scripts shared on Dataverse. They found that 75% of R scripts cannot be successfully executed. Among these failed R scripts is an R script shared by Beck [17].

This R script has been "rescued" by the author of the R package `groundhog` [13], as demonstrated in a blog post (http://datacolada.org/100). We were wondering if `rang` can also be used to "rescue" the concerned R script. The date of the R script, as indicated on Dataverse, is 2018-12-12. This date is used as the snapshot date.

```
## as_pkgrefs is automatically run in this case
graph <- resolve("nathaniel", snapshot_date = "2018-12-12")
dockerize(graph, output_dir = "nat", materials_dir = "nathaniel")

cd nat
docker build -t nat .
docker run --rm --name "natcontainer" --entrypoint bash -ti nat
```

Inside the container

```
cd materials
Rscript fn_5.R
```

The same file can thus also be "rescued" by `rang`.

## Recover a removed R package: maxent

The R package `maxent` introduces a machine learning algorithm with a small memory footprint and was available on CRAN until 2019. A software paper was published by the original authors in 2012 [18]. The R package was also used in some subsequent automated content analytic papers [e.g. 19]. Despite the editing of the package by a staffer of CRAN, the package was removed from CRAN in 2019. (Here, we said the package `maxent` was "removed from CRAN" as per what the webpage (https://cran.r-project.org/web/packages/maxent/index. html) is written. However, the terminology as stated in the CRAN Policies is "archived", quote: "Packages will not normally be removed from CRAN: however, they may be archived, including at the maintainer's request." As all CRAN packages have all the submitted versions archived, we find this terminology confusing. Therefore, we use "removed" packages throughout this paper to indicate packages that cannot be installed by the usual method, i.e. `install.packages` but still have the old versions archived on CRAN.) We attempted to install the second last (the original submitted version) and last (with editing) versions of `maxent` on R 4.2.2. Neither worked.

Using `rang`, we are able to reconstruct a computational environment with R 2.15.0 (2012-03-30) to run all code snippets published in Jurka [18]. For removed CRAN packages, we strongly recommend querying the Github read-only mirror of CRAN instead (https://github.com/cran). This way, the resolved system requirements have a higher chance of being correct.

```
maxent <- resolve("cran/maxent", "2012-06-10")
dockerize(maxent, "maxentdir", cache = TRUE)
```

## Recover a removed R package: ptproc

The software paper of the R package `ptproc` was published in 2003 and introduced multi-dimensional point process models [20]. But the package has been removed from CRAN for over a decade (at least). The only release on CRAN was on 2002-10-10. The package is still listed in the "Handling and Analyzing Spatio-Temporal Data" CRAN Task View despite being uninstallable without modification on any modern R system (see below). As of writing the package is still downloadable from the original author's website as a tarball file (tar.gz).

Even with this over-a-decade removal and new packages with similar functionalities have been created, there is evidence that `ptproc` is still being sought for. As late as 2017, there are blog posts on how to install the long obsolete package on modern versions of R. The package is extremely challenging to install on a modern R system because it was written before the introduction of name space management in R 1.7.0 [21]. In other words, the available tarball files from the original author's website and CRAN do not contain a `NAMESPACE` file like all other modern R packages do.

The oldest version of R that `rang` can support, as of writing, is R 1.3.1. `rang` is probably the only solution available that can support the 1.x series of R (i.e. before 2004-10-04). Similar to the case of `maxent` above, a `Dockerfile` to assemble a Docker image with `ptproc` installed can be generated with two lines of code.

```
graph <- resolve("ptproc", snapshot_date = "2004-07-01")
dockerize(graph, "~/dev/misc/ptproc", cache = TRUE)
```

Suppose we have an R script, extracted from Peng [20], called "peng.R" like this:

```
library(ptproc)

set.seed(1000)
x <- cbind(runif(100), runif(100), runif(100))
hPois.cond.int <- function(params, eval.pts, pts = NULL,
                           data = NULL, TT = NULL) {
    mu <- params[1]
    if(is.null(TT))
        rep(mu, nrow(eval.pts))
    else {
        vol <- prod(apply(TT, 2, diff))
        mu * vol
    }
}
ppm <- ptproc(pts = x, cond.int = hPois.cond.int, params = 50,
              ranges = cbind(c(0,1), c(0,1), c(0,1)))
fit <- ptproc.fit(ppm, optim.control = list(trace = 2), method = "BFGS")
summary(fit)
```

One can integrate `rang` into a BASH script to completely automate the batch execution of the above R script.

```
Rscript -e "library(rang); dockerize(resolve('ptproc', '2004-07-01'),
'pengdocker', cache = TRUE)"
docker build -t pengimg ./pengdocker
## launching a container in daemon mode -d
docker run -d --rm --name "pengcontainer" -ti pengimg
docker cp peng.R pengcontainer:/peng.R
docker exec pengcontainer R CMD BATCH peng.R
docker exec pengcontainer cat peng.Rout
docker cp pengcontainer:/peng.Rout peng.Rout
docker stop pengcontainer
```

The file peng.Rout contains the execution results of the script from inside the Docker container. As the random seed was preserved by the original author [20], the above BASH script can perfectly reproduce the analysis. It is also important to note that the random number generator (RNG) of R has been changed several times over the course of the development. In this case, we are using the same generation of RNG as Peng [20] in order to reproduce the analysis.

### Recover a removed Bioconductor package

Similar to CRAN, packages can also be removed over time from Bioconductor. The Bioconductor package Sushi has been deprecated by the original authors and is removed from Bioconductor version 3.16 (2022-11-02). Sushi is a data visualization tool for genomic data and was used in many online tutorials and scientific papers, including the original paper announcing the package by the original authors [22].

rang has native support for Bioconductor packages since version 0.2. We obtained the R script "PaperFigure.R" from the Github repository of Sushi, which generates the figure in the original paper [22]. Similar to the above case of ptproc, we made a completely automated BASH script to run "PaperFigure.R" and get the generated figure out of the container (Fig 2). We made no modification to "PaperFigure.R".

```
Rscript -e "library(rang); dockerize(resolve('Sushi', '2014-06-05'),
'sushidocker', no_rocker = TRUE, cache = TRUE)"
docker build -t sushiimg ./sushidocker
docker run -d --rm --name "sushicontainer" -ti sushiimg
docker cp PaperFigure.R sushicontainer:/PaperFigure.R
docker exec sushicontainer mkdir vignettes
docker exec sushicontainer R CMD BATCH PaperFigure.R
docker cp sushicontainer:/vignettes/Figure_1.pdf sushi_figure1.pdf
docker stop sushicontainer
```

## Preparing executable compendia with long-term computational reproducibility

The above six examples show how powerful rang is to reconstruct tricky computational environments which have not been completely declared in the literature. Although we position rang mostly as an archaeological tool, we think that rang can also be used to prepare executable research compendia of current research. We cannot predict the future but research compendia generated by rang would probably have long-term computational reproducibility.

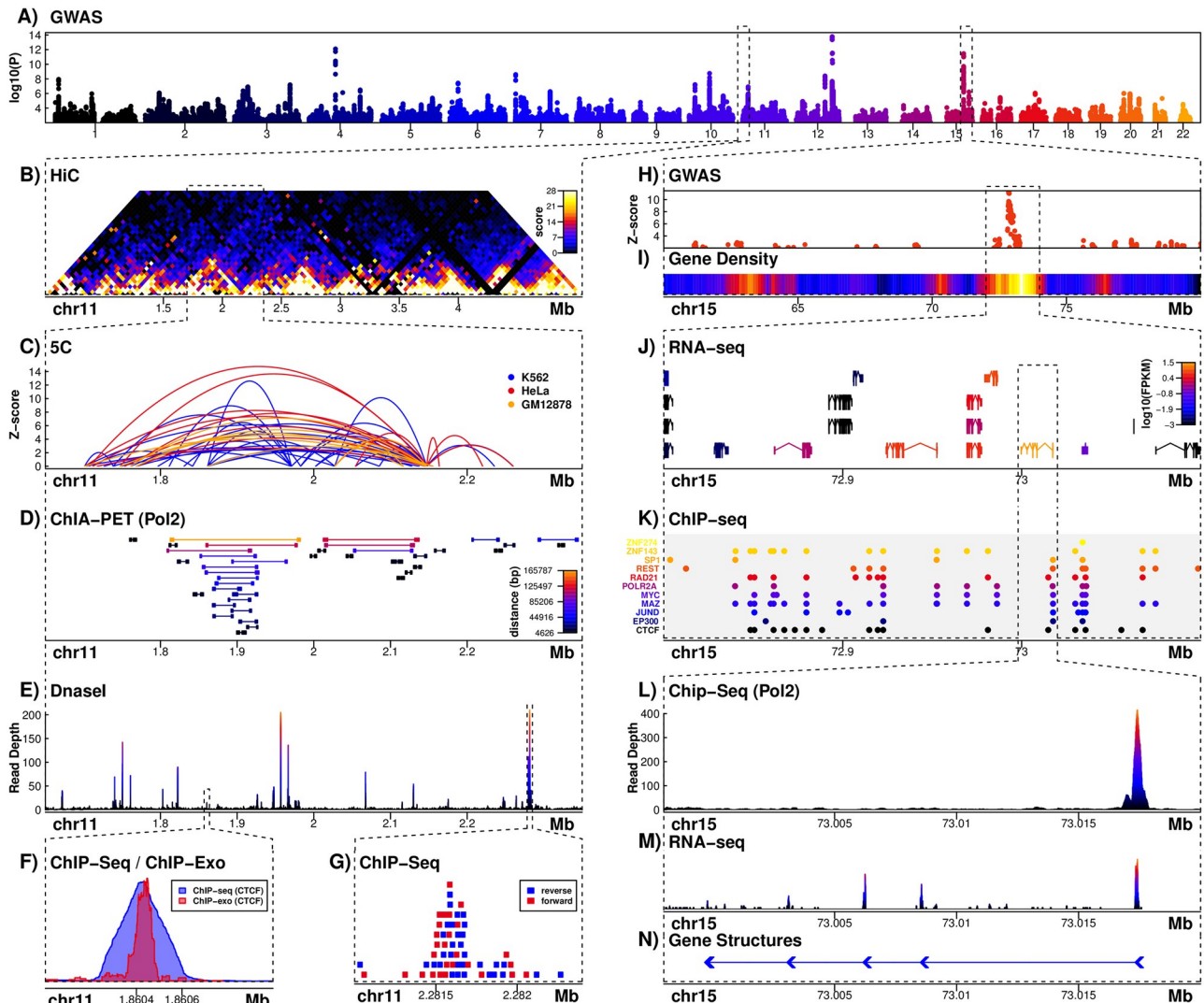

**Fig 2. The figure from the batch execution of PaperFigure.R inside a Docker container generated by rang.**

To demonstrate this point, we took the recent paper by Oser et al. [23]. This paper was selected because 1) the paper was published in *Political Communication*, a high impact journal that awards Open Science Badges; 2) shared data and R code are available; and most importantly, 3) the shared R code is well-written. In the repository of this paper, we based on the materials shared by Oser et al. [23] and prepared a research compendium that should have long-term computational reproducibility. The research compendium is similar to the Executable Compendium suggested by the Turing way [24].

The preparation of the research compendium is easy since `rang` can scan a materials directory for all R packages used. We detected a minor issue in the code base that an undeclared Github package is used. But it can be easily solved, as in the Psychological Science example above.

```
library(rang)
## meta-analysis is the directory of all shared materials
cran_pkgs <- as_pkgrefs("meta-analysis")

## dmetar is an undeclared github package: MathiasHarrer/dmetar
cran_pkgs[cran_pkgs == "cran::dmetar"] <- "MathiasHarrer/dmetar"
x <- resolve(cran_pkgs, "2021-08-11", verbose = TRUE)
##print(x, all_pkgs = TRUE)
dockerize(x, "oserdocker", materials_dir = "meta-analysis", cache = TRUE)
```

The above R script is saved as `oser.R`. The central piece of the executable compendium is the `Makefile` [25].

```
output_file=reproduced.html
r_cmd = "rmarkdown::render('materials/README.Rmd',\
output_file = '${output_file}')"
handle=oser
local_file=${handle}_README.html

.PHONY: all resolve build render export rebuild

all: resolve build render
    echo "finished"
resolve:
    Rscript ${handle}.R
build: ${handle}docker
    docker build -t ${handle}img ${handle}docker
render:
    docker run -d --rm --name "${handle}container" -ti ${handle}img
    docker exec ${handle}container Rscript -e ${r_cmd}
    docker cp ${handle}container:/materials/${output_file} ${local_-
file}
    docker stop ${handle}container
export:
    docker save ${handle}img | gzip > ${handle}img.tar.gz
rebuild: ${handle}img.tar.gz
    docker load < ${handle}img.tar.gz
```

With this `Makefile`, one can create the `Dockerfile` with `make resolve`, build the Docker image with `make build`, render the RMarkdown file inside the container with `make render`, export the built Docker image with `make export`, and rebuild the exported Docker image with `make rebuild`.

The structure of the entire executable compendium looks like this:

```
Makefile
oser.R
meta-analysis/
README.md
oserdocker/
oserimg.tar.gz
```

In this executable compendium, only the first four elements are essential. The directory `oserdocker` (116 MB) contains cached R packages, a `Dockerfile`, and a verbatim copy of the directory `meta-analysis/` to be transferred into the Docker image. That can be regenerated by running `make resolve`. However, preserving this directory insures against

the situations where (1) some R packages used in the project are no longer available, (2) any of the information providers used by `rang` for resolving the dependency relationships are not available, or (3) in the rare circumstance of `rang` is no longer available.

`oserimg.tar.gz` (667 MB) is a backup copy of the Docker image. This can be regenerated by running `make export`. Preserving this file insures against all the situations mentioned above, but also the situations of Docker Hub (the hosting service provided by Docker for base images such as Rocker) and the software repositories used by the dockerized operating system being not available. When `oserimg.tar.gz` is available, it is possible to run `make rebuild` and `make render` even without internet access (provided that Docker and `make` have been installed before). Of course, there is still an extremely rare situation where Docker (the program) itself is no longer available. However, it is possible to convert the image file for use on other containerization solutions such as Singularity, if Docker is really not available anymore.

Sharing of research artifacts less than 1G is not as challenging as it used to be. Zenodo, for example, allows the sharing of 50G of files. Therefore, sharing of the last two components of the executable compendium prepared with `rang` is at least possible on Zenodo (https://doi.org/10.5281/zenodo.7708417). However, for data repositories with more restrictions on data size, sharing the executable compendium without the last two parts could be considered sufficient. For that, run `make` will make the default target `all` and generate all the things needed for reproducing the analysis inside a container.

The above `Makefile` is general enough that one can reuse it by just modifying how the R scripts (the `r_cmd` variable) in the `materials` directory are executed. This can be a starting point of a standard executable compendium format.

## Concluding remarks

This paper presents `rang`, a solution to (re)construct R computational environments based on Docker. As the six examples in Section 3 show, `rang` can be used archaeologically to rerun old code, many of them not executable without the analytic and reconstruction processes facilitated by `rang`. These retrospective use cases demonstrate how versatile `rang` is. `rang` is also helpful for prospective usage, as demonstrated in the section where an executable compendium is introduced.

There are still many features that we did not cover in this paper (see Table 1). `rang` is built with interoperability in mind. As of writing, `rang` is interoperable with existing R packages such as `renv` and R built-in `sessionInfo()`. Also, the `rang` object can be used for network analysis with R packages such as `igraph`.

Computational reproducibility is a complex topic and as in all of these complex topics, there is no silver bullet [26]. All solutions have their trade-offs and cannot solve all issues.

**Table 1. List of all functions in rang version 0.2.0.**

| Function | Purpose |
| --- | --- |
| `as_pkgrefs` | Convert data structures (e.g. `sessionInfo`) into package references |
| `convert_edgelist` | Convert the resolved results to edgelist |
| `dockerize` | Dockerize the resolved result |
| `export_rang` | Export the resolved result as installation script |
| `export_renv` | Export the resolved result as renv lockfile |
| `query_sysreqs` | Query for system requirements |
| `resolve` | Resolve dependencies of R packages |

Although `rang` can restore a specific computational environment, there might be potential barriers to reproducibility related to external dependencies which cannot be containerized. A common external dependency is a data source connected via a web application programming interface (API). Some examples are so-called "API packages" such as `STRINGdb` (an R interface to the STRING protein-protein database), `rtoot` (an R interface to the Mastodon API), and `googleLanguageR` (an R interface to the Google Translation API). Although these API packages can be containerized, the external data sources cannot. Therefore, `rang` is not a solution to issues related to external dependencies. For data collection via these API packages, we recommend caching the collected data and sharing the cached data. For computation via API packages (e.g. Google Translate), we recommend using alternative methods that do not require external dependencies (e.g. [27]).

Moreover, the (re)construction process based on `rang` takes notably more time than other solutions because all packages are compiled from source. `rang` trades computational efficiency of this often one-off (re)constructing process for correctness, backward compatibility and independence from any commercial backups of software repositories such as MRAN. In the Vignette of `rang` (https://cran.r-project.org/web/packages/rang/vignettes/faq.html), we list all limitations: (1) `rang` does not support R < 1.3.1, (2) `rang` can generate Ubuntu/Debian-based Docker images and therefore non-Linux features (e.g. WinBUGS) are not supported, (3) `dockerize()` cannot cache OS-level packages (`deb` packages), (4) R packages with changing system requirements might be incorrectly queried, (5) caching of R packages and R source must be enforced when the R version is too old, (6) R packages on CRAN, Bioconductor, and Github, as well as Rocker images, might not be available in the future. All caveats listed have no solution, except point (6) can be mitigated by backing up the generated Docker image (see the section on executable compendia).

## Author Contributions

**Conceptualization:** Chung-hong Chan, David Schoch.

**Investigation:** Chung-hong Chan, David Schoch.

**Methodology:** Chung-hong Chan, David Schoch.

**Project administration:** David Schoch.

**Software:** Chung-hong Chan, David Schoch.

**Supervision:** David Schoch.

**Visualization:** Chung-hong Chan.

**Writing – original draft:** Chung-hong Chan, David Schoch.

**Writing – review & editing:** Chung-hong Chan, David Schoch.

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
