## [Decision Letter · Decision Letter 0]

17 Apr 2023

PONE-D-23-07706rang: Reconstructing reproducible R computational environmentsPLOS ONE

Dear Dr. Chan,

Thank you for submitting your manuscript to PLOS ONE. After careful consideration, we feel that it has merit but does not fully meet PLOS ONE’s publication criteria as it currently stands. Therefore, we invite you to submit a revised version of the manuscript that addresses the points raised during the review process.

We look forward to receiving your revised manuscript.

Kind regards,

Carlos Fernandez-Lozano, Ph.D

Academic Editor

PLOS ONE

Journal Requirements:

Additional Editor Comments (if provided):

Please address the reviewer's comments to improve your outstanding manuscript.

Reviewers' comments:

Reviewer's Responses to Questions

**Comments to the Author**

1. Is the manuscript technically sound, and do the data support the conclusions?

Reviewer #1: Yes

Reviewer #2: Yes

2. Has the statistical analysis been performed appropriately and rigorously? 

Reviewer #1: N/A

Reviewer #2: Yes

3. Have the authors made all data underlying the findings in their manuscript fully available?

Reviewer #1: Yes

Reviewer #2: Yes

4. Is the manuscript presented in an intelligible fashion and written in standard English?

Reviewer #1: Yes

Reviewer #2: Yes

5. Review Comments to the Author

Reviewer #1: General comments

This manuscript describes the ‘rang’ package for R, focusing on six examples of using rang to reconstruct the computational environments of old scripts. This is important and interesting, for reasons well described in the introduction—basically, computational reproducibility is a hard problem, and while not a panacea this package provides useful functionality to help this. The ms is fairly well written, and the core examples are very well done in their variety and step-by-step explanations.

There are some minor problems (see below), and one major one I thought: if the purpose of this ms is to describe the rang package, i.e. to be its primary peer-reviewed documentation, it needs to list all package functions and what they do (e.g. in a table; see #12 below).

In summary, this is an interesting and well-done ms that usefully lays out how to use rang to reconstruct computational environments in a wide variety of cases. It needs minor to moderate revisions for clarity in many places, but is fundamentally a strong piece worthy of publication.

Specific comments

1. Abstract: could probably say “usually missing”

2. Abstract: change “spanning from” to “spanning” (grammar)

3. Line 13: cite R correctly – see citation()

4. L. 24: this is a bit odd, as slurm is a workload scheduler, not a computing environment

5. L. 64: “unambiguously specify” instead of “pin down”? Seems clearer

6. L. 75: what limitations, exactly? Be specific

7. L. 159: a little unclear. “which is used by the scanning function”?

8. L. 186: “covert editing”? Really? Clarify, expand, or remove

9. L. 189: “Neither worked.”

10. L. 245: what does “suggested by the Turing way” mean?

11. L. 248: use “library(rang)” (cf. code on p. 3) not “require(rang)”

12. L. 291: there are features not mentioned? Like what? A table listing all the package functions would be a useful addition

13. The major rang caveats, as listed in the package vignette, should be included in this article

Reviewer #2: The article is well written and the package works as described. They have tested rang using a wide range of examples and the steps are well documented and clear. The problem they are addressing is a very important one and I commend their effort in creating the package and producing the article.

My only comment is related to the discussion of limitations. Many packages access databases to obtain data. For example, the STRINGdb package has a function to obtain protein interaction data, and the API changes over time. As such, previous versions of STRINGdb cannot access the db. For packages like biomaRt, the method to access the data may be more stable, however the actual data itself can change, for example Gene Ontology data. I suggest the authors make this limitation clear and/or suggest ways to solve, or in some way minimise it.

I would also recommend a thorough language edit - the article is very clearly written and easy to follow but there are some small errors. For example, in the following sentence, there is some mixing of tenses, and the use of brackets seems unsual:

However, having this

directory preserved insures against the situations that some R packages used in the

project were no longer available or any of the information providers used by rang for

resolving the dependency relationships were not available. (Or in the rare circumstance

of rang is no longer available.)

6. PLOS authors have the option to publish the peer review history of their article (what does this mean?). If published, this will include your full peer review and any attached files.

Reviewer #1: **Yes: **Ben Bond-Lamberty

Reviewer #2: No

---

## [Author Response · Author response to Decision Letter 0]

22 May 2023

# Reviewer #1: General comments

**R1.1: This manuscript describes the ‘rang’ package for R, focusing on six examples of using rang to reconstruct the computational environments of old scripts. This is important and interesting, for reasons well described in the introduction—basically, computational reproducibility is a hard problem, and while not a panacea this package provides useful functionality to help this. The ms is fairly well written, and the core examples are very well done in their variety and step-by-step explanations.**

A1.1 We would like to thank R1 for their appreciation of our package `rang`.

**R1.2: There are some minor problems (see below), and one major one I thought: if the purpose of this ms is to describe the rang package, i.e. to be its primary peer-reviewed documentation, it needs to list all package functions and what they do (e.g. in a table; see #12 below).**

A1.2 We agreed with R1 that a list of all package functions is needed. It has been added accordingly. (See table 1)

**R1.3: In summary, this is an interesting and well-done ms that usefully lays out how to use rang to reconstruct computational environments in a wide variety of cases. It needs minor to moderate revisions for clarity in many places, but is fundamentally a strong piece worthy of publication.**

A1.3 We would like to thank R1 for their appreciation of our paper.

**R1.4 Specific comments 1. Abstract: could probably say “usually missing”**

A1.4 We adopted the suggestion by R1.

**R1.5 2. Abstract: change “spanning from” to “spanning” (grammar)**

A1.5 We adopted the suggestion by R1.

**R1.6 3. Line 13: cite R correctly – see citation()**

A1.6 We cited R accordingly.

**R1.7 4. L. 24: this is a bit odd, as slurm is a workload scheduler, not a computing environment**

A1.7 R1 correctly pointed out the problem. We removed the mention of slurm.

**R1.8 5. L. 64: “unambiguously specify” instead of “pin down”? Seems clearer**

A1.8 We adopted the R1's suggestion to say "(unambigously) specify" instead of "pin down". And we agree that it is clearer.

**R1.9 6. L. 75: what limitations, exactly? Be specific**

A1.9 We spelled out the limitations of MRAN.

**R1.10 7. L. 159: a little unclear. “which is used by the scanning function”?**

A1.10 We stated clearly that `as_pkgrefs()` is a wrapper.

**R1.11 8. L. 186: “covert editing”? Really? Clarify, expand, or remove**

A1.11 As far as we know, the editing was done without the permission from the original submitter. However, without going too detail into similar editing by the same staffer, we decided to drop the adjective "covert". 

**R1.12 9. L. 189: “Neither worked.”**

A1.12 We adopted the suggestion by the reviewer.

**R1.13 10. L. 245: what does “suggested by the Turing way” mean?**

A1.13 The Turing Way (https://the-turing-way.netlify.app/reproducible-research/compendia.html) is a handbook published by the Alan Turing Institute in the UK. We provided a citation to the handbook to make it clearer.

**R1.14 11. L. 248: use “library(rang)” (cf. code on p. 3) not “require(rang)”**

A1.14 We adopted the suggestion by the reviewer.

**R1.15 12. L. 291: there are features not mentioned? Like what? A table listing all the package functions would be a useful addition**

A1.15 See A1.2

**R1.16 13. The major rang caveats, as listed in the package vignette, should be included in this article**

A1.16 We adopted the suggestion by the reviewer to include all caveats.

# Reviewer #2:

**R2.1 The article is well written and the package works as described. They have tested rang using a wide range of examples and the steps are well documented and clear. The problem they are addressing is a very important one and I commend their effort in creating the package and producing the article.**

A2.1 We would like to thank R2 for their appreciation of our package `rang` and the article.

**R2.2 My only comment is related to the discussion of limitations. Many packages access databases to obtain data. For example, the STRINGdb package has a function to obtain protein interaction data, and the API changes over time. As such, previous versions of STRINGdb cannot access the db. For packages like biomaRt, the method to access the data may be more stable, however the actual data itself can change, for example Gene Ontology data. I suggest the authors make this limitation clear and/or suggest ways to solve, or in some way minimise it.**

A2.2 We agree with R2 that this is a major reproducibility issue. In the revised version of the paper, we added this as a limitation. `rang` is not a solution to these external dependencies and we made several suggestions.

**R2.3 I would also recommend a thorough language edit - the article is very clearly written and easy to follow but there are some small errors. For example, in the following sentence, there is some mixing of tenses, and the use of brackets seems unsual: However, having this directory preserved insures against the situations that some R packages used in the project were no longer available or any of the information providers used by rang for resolving the dependency relationships were not available. (Or in the rare circumstance of rang is no longer available.)**

A2.3 We revised the language of the paper. The sentence mentioned was due to the wrong usage of past subjunctive mood ("were"). We fixed it to use present indicative mood instead.

---

## [Editor Report · Decision Letter 1]

23 May 2023

rang: Reconstructing reproducible R computational environments

PONE-D-23-07706R1

Dear Dr. Chan,

We’re pleased to inform you that your manuscript has been judged scientifically suitable for publication and will be formally accepted for publication once it meets all outstanding technical requirements.

Kind regards,

Carlos Fernandez-Lozano, Ph.D

Academic Editor

PLOS ONE

Additional Editor Comments (optional):

I am writing to extend my warmest congratulations on the publication of your manuscript. This is truly a remarkable achievement and a testament to your dedication and hard work.
---

## [Editor Report · Acceptance letter]

30 May 2023

PONE-D-23-07706R1 

rang: Reconstructing reproducible R computational environments  

Dear Dr. Chan:

I'm pleased to inform you that your manuscript has been deemed suitable for publication in PLOS ONE. Congratulations! Your manuscript is now with our production department. 

Kind regards, 

on behalf of

Dr. Carlos Fernandez-Lozano 

Academic Editor

PLOS ONE